# Association of Lifestyle Factors and Neuropsychological Development of 4-Year-Old Children

**DOI:** 10.3390/ijerph17165668

**Published:** 2020-08-05

**Authors:** Giselle O’Connor, Jordi Julvez, Silvia Fernandez-Barrés, Eva Mᵃ Navarrete-Muñoz, Mario Murcia, Adonina Tardón, Isolina Riaño Galán, Pilar Amiano, Jesús Ibarluzea, Raquel Garcia-Esteban, Martine Vrijheid, Jordi Sunyer, Dora Romaguera

**Affiliations:** 1ISGlobal, Instituto de Salud Global de Barcelona-Campus MAR, 08003 Barcelona, Spain; giselle.m.oconnor@gmail.com (G.O.); silvia.fernandez@isglobal.org (S.F.-B.); raquel.garcia@isglobal.org (R.G.-E.); martine.vrijheid@isglobal.org (M.V.); jordi.sunyer@isglobal.org (J.S.); dora.romaguera@isglobal.org (D.R.); 2Institut d’Investigació Sanitària Pere Virgili (IISPV), Hospital Universitari Sant Joan de Reus, 43204 Reus, Spain; 3Centro de Investigacion Biomedica en Red Epidemiologia y Salud Pública (CIBERESP), 28029 Madrid, Spain; enavarrete@umh.es (E.M.N.-M.); murcia_mar@gva.es (M.M.); atardon@uniovi.es (A.T.); isolinariano@gmail.com (I.R.G.); mambien3-san@euskadi.eus (J.I.); 4Nutritional Epidemiology Unit Department of Public Health, History of Medicine and Gynecology, Universidad Miguel Hernández, 03550 Alicante, Spain; 5Grupo de Investigación en Terapia Ocupacional (InTeO), Miguel Hernández University, 03550 Alicante, Spain; 6Epidemiology and Environmental Health Joint Research Unit, FISABIO-Universitat Jaume I-Universitat de València, 08034 Valencia, Spain; 7Department of Medicine, University of Oviedo, 33003 Oviedo, Spain; 8Hospital Universitario Central de Asturias, 33003 Oviedo, Spain; 9Public Health Division of Gipuzkoa, BioDonostia Health Research Institute, 20014 San Sebastian, Spain; epicss-san@euskadi.eus; 10Faculty of Psychology, University of the Basque Country (UPV/EHU), 20014 San Sebastian, Spain; 11Department of Experimental and Health Sciences, Universitat Pompeu Fabra (UPF), 08002 Barcelona, Spain; 12Institut D’Investigació Sanitària Illes Balears (IdISBa), 07010 Palma de Mallorca, Spain; 13Centro de Investigación Biomédica en Red de Fisiopatología de la Obesidad y Nutrición (CIBEROBN), 28029 Madrid, Spain

**Keywords:** cohort study, neuropsychological development, lifestyle factors, child health

## Abstract

Background: We aimed to assess how lifestyle factors such as diet, sleep, screen viewing, and physical activity, individually, as well as in a combined score, were associated with neuropsychological development in pre-school age children. Methods: We conducted a cross-sectional study in 1650 children of 4 years of age, from the Environment and Childhood Project (INMA) population-based birth cohorts in four regions of Spain. Children were classified per a childhood healthy lifestyle score (CHLS) with a range of 0 to 4 that included eating in concordance with the Mediterranean diet (1 point); reaching recommended sleep time (1 point); watching a maximum recommended screen time (1 point); and being physically active (1 point). The McCarthy Scales of Children’s Abilities (MSCA) were used to test neuropsychological development. Multi-adjusted linear regression models were created to assess the association with the lifestyle factors individually and as a combined score. Results: CHLS was not associated with MSCA general cognitive score (1-point increment = −0.5, 95% CI: −1.2, 0.2). Analyzed by separate lifestyle factors, physical activity had a significant negative association with MSCA score and less TV/screen time had a negative association with MSCA score. Conclusion: In this cross-sectional study, a combined score of lifestyle factors is not related to neuropsychological development at pre-school age.

## 1. Introduction

Unhealthy lifestyle factors make an important contribution to the modifiable disease burden and are one of the largest concerns for public health [1]. A child’s unhealthy lifestyle can affect development, as well as continue through life, while their health and the population’s health decline. Child lifestyle factors like adequate sleep time, a healthy diet, appropriately limited screen-viewing time, and sufficient physical activity have all previously been positively related to neurodevelopmental outcomes [2,3,4,5,6,7,8,9,10,11]. Having an unhealthy lifestyle could potentially damage neuropsychological development and alter cognitive function. Our interest in neuropsychological development stems from the concern that unhealthy habits leave children fundamentally disadvantaged from the beginning of life.

A lack of sleep has been found to be detrimental to certain neurocognitive functions and is associated with more behavioral problems [4]. While studies on pre-school age children are limited, studies suggest that children who sleep more have better results on a variety of cognitive tests [12]. More screen viewing has been loosely shown to be detrimental to young children, specifically by taking away sleep time. Studies have also found that the type of screen time is important to consider, with child-specific television content being beneficial in some studies [3,4,5,6,7].

There has been limited research on physical activity, since young children have historically been assumed to be active, and research suggests that higher amounts of physical activity in children are associated with better neurocognitive development [2,3]. There is strong evidence for physical activity and sports being beneficial for cognitive function by supporting brain structure and function in adolescence and pre-adolescence, with weak evidence in younger pre-school age children that physical activity is beneficial to cognitive health [13,14,15]. A systematic review of physical activity and diet found that, while most studies showed that acute and long-term physical activity exposure was positively associated with neurocognitive development, most of the studies were limited in the number of participants and offered weak evidence [8]. Conversely, sedentary behaviors, such as reading, were found to be beneficial to cognitive development [3].

Diet quality has been associated with better neurocognitive development in children; healthy dietary patterns, such as the Mediterranean diet, have been associated with less cognitive decline in adults and healthy diets have been associated with better reading comprehension and fluency in grade school children [11,16,17]. However, some results were null and many were strongly attenuated when controlling for factors such as socioeconomic status.

The aim of the current study was to assess how lifestyle factors, such as diet, sleep, screen viewing, and physical activity, both assessed together using an index score and individually, are associated with neuropsychological development in 4-year-old children in the Environment and Childhood Project (INMA) birth cohort study. While many previous studies have looked at lifestyle factors as individual contributors to neuropsychological activity, few have combined the lifestyle factors into an encompassing score reflecting on wellbeing. Looking at a cross-sectional study of children with a combined score of lifestyle factors, and by individual factors, we hope to best represent the reality of daily life.

## 2. Methods

### 2.1. Study Population and Data Collection

The Environment and Childhood Project (INMA) is a multi-center Spanish cohort study with the objective of examining the health effects of early life exposures on child health [18]. In our analysis, pregnant women from four of the cohorts in the cities of Sabadell, Asturias, Gipuzkoa, and Valencia were recruited at a prenatal check-up if they met the inclusion criteria of (a) being a resident in the respective cohort area, (b) being at least 16 years old, (c) having a singleton pregnancy, (d) not having followed any assisted reproduction program, (e) wishing to deliver at the cohort’s reference hospital, and (f) not having communication problems. The appropriate ethical committees for each center approved the study and written informed consent was obtained from all adult participants before enrolment in the study and in each follow up. All subjects gave their informed consent for inclusion before they participated in the study. The study was conducted in accordance with the Declaration of Helsinki, and the protocol was approved by the the regional ethical committees of each cohort (project identification code: 2009/3432/l). Mother–child pairs were afterwards followed up periodically. At the 4-year follow-up, between 2008 and 2012, parents responded to a questionnaire pertaining to their child’s lifestyle. Children were assessed through a series of physical exams and cognitive tests. We started with a sample of 2644 participants and excluded participants without neuropsychological testing results or acceptable testing quality and those who did not have the exposure and confounding measurements noted in Table 1, leaving a final sample size of 1650.

### 2.2. Exposure Variables

Child diet information was collected through a validated 105-item semi-quantitative food frequency questionnaire (FFQ) given to parents in the INMA study [19]. A healthy dietary pattern was assessed by means of an a priori index, the Alternate Mediterranean Diet score (aMED) [20]. The consumption of vegetables, fruits, nuts, whole grain cereals, legumes, fish, the ratio of monounsaturated fatty acids divided by saturated fatty acids, and total red and processed meat products were measured in servings or nutrients per day. Children consuming above the median intake of vegetables, fruits, nuts, whole grain, cereals, legumes, fish, and ratio of fatty acids got 1 point (and 0 otherwise); for red and processed meat, children consuming below the median got 1 point, and 0 otherwise. The scores for each food group were summed, and the final aMED range was 0–8. Children with an aMED score greater than or equal to than 4 were considered to have healthier dietary patterns and scored one point in the lifestyle score.

Sleep time information was collected by interview by asking how long children slept daily at night and during naps. The variable was dichotomized using recommended cut offs for average sleep time in preschool age children. The lower limit threshold for a healthy sleep amount was set at 10 h of sleep a day in accordance with current recommendations [21]. Children sleeping 10 h or more per day scored one point.

Television viewing and other screen time information was collected by asking parents how many hours per week children watched television or videos, and was dichotomized to consider unhealthy television viewing to be more than 1 h per day, using the official recommendations at the time from the American Academy of Pediatrics [22]. Children with 1 h or less per day of television viewing scored one point.

Physical activity data were collected through a population-validated survey [23] by asking parents how much time their child spent in organized or non-organized physical activities outside of school. The variable was dichotomized with the healthy choice being more than the median number of Metabolic Equivalents (METs) in each respective cohort. We decided that the large variance between cohorts called for a cohort-specific median-centered physical activity variable. Children with physical activity above the median scored one point.

We created a childhood healthy lifestyle score (CHLS) using the dichotomous variables for diet, sleep, screen time, and physical activity. The final summation of these variables was used as the main exposure variable. The total CHLS was categorized, with one point for each healthy lifestyle factor; hence the score could have had a value between 0 (least healthy lifestyle) and 4 (healthiest lifestyle). The CHLS was further categorized in four categories (scores of 0 and 1 were combined and used as the reference category; 2; 3; and 4 points).

### 2.3. Outcome Variable

The studied outcome was the cognitive development of child at 4 years old, assessed using the Spanish version of the McCarthy Scales of Children’s Abilities (MSCA), administered by trained psychologists [24]. The combined score of the general cognitive index was calculated by utilizing the verbal and perceptual performance scales and numerical scales. Furthermore, Cronbach’s alpha coefficient was used to determine the internal consistency per each of the scales [25]. A good coefficient would be a value of ≥0.70, scale alpha coefficients were >0.70.

### 2.4. Covariates

Parents completed questionnaires and interviews during prenatal visits. We obtained data on maternal and paternal educational level (primary or lower, secondary, university degree). We collected maternal and paternal social class information in a self-reported questionnaire, organized based on Goldthorpe’s criteria for social economic class using the occupation at the time of the pregnancy as socioeconomic status (I–II, managers/technicians; III, skilled; IV–VI, semiskilled/unskilled) [26]. Exposure to tobacco smoking at 4 years old was assessed in the parental survey. We obtained a proxy score of maternal verbal Intelligence Quotient (IQ) by the Wechsler Adult Intelligence Scales III similarities subtest [27]. Trained staff measured child weight (nearest gram) and height (nearest 0.1 cm) at 4 years old using standard protocols. We calculated child body mass index (BMI, weight/length^2^) and estimated age- and sex-specific BMI z-scores based on the WHO referent [28].

### 2.5. Statistical Analysis

A descriptive analysis was done, presenting means and standard deviations for quantitative variables, and the sample number and the percentage in each category for qualitative variables hypothesized to be relevant a priori. Differences in categorical variables were analyzed with a chi-squared test and an ANOVA test was used for continuous variables.

Multi-adjusted linear regression models were created to estimate the change in coefficient in MSCA associated with the lifestyle factors at 4 years of age. Firstly, associations between each lifestyle factors individually and MSCA score were modeled using three levels of adjustment: model 1 controlled for variables that were considered essential a priori (quality of neurological test, age at testing, sex, and cohort); model 2 was further adjusted by covariates that affected both the exposure and the outcome variable tested by additive model building (*p* < 0.2): maternal and paternal socioeconomic class and child’s exposure to tobacco smoke; model 3 was mutually adjusted for all the other lifestyle variables. Covariates that were not adjusted for, although they tested as significant, were BMI categories, maternal education, and maternal IQ. There was a large percentage of missing data for maternal IQ and, given that maternal social class status is correlated with maternal IQ, we decided to only adjust for maternal social class. We decided to exclude maternal education for a similar reason, since we adjusted the models for maternal social class. However, we adjusted for all of them in sensitivity analyses. Furthermore, BMI was selected as part of the sensitivity analyses only, given that BMI was thought to be a mediator of the association between lifestyle and neurodevelopment. We additionally built a Directed Acyclic Graph (DAG) model in order to help us with the confounding variable selection for the final models (Appendix A).

The association between the CHLS in four categories (0–1 points, 2, 3, and 4 points) and the MSCA score was modeled with two different levels of adjustment: model 1 and model 2, with the same covariates as explained above. The lifestyle factors were analyzed for correlation and collinearity between factors and were not found to have sufficient correlation or collinearity to warrant a different analysis (Appendix A).

In light of the initial results, we stratified both models by maternal social class into two categories: manual laboring classes (classes IV–VI) versus non-manual workers (classes I+II and III combined), and we tested the interaction by social class using a Wald test. As a further sensitivity analysis, we excluded children with diagnosed attention deficit/hyperactivity disorder symptoms (*n* = 87). Finally, we stratified by sex and cohort and compared children included and excluded in the analyses.

All statistical analyses were done using STATA version 12 (STATA, College Station, TX, USA) and statistical significance was defined as having a *p*-value < 0.05.

## 3. Results

Data collected from Sabadell, Asturias, Gipuzkoa, and Valencia (Spain) had populations with an equal distribution of sexes. Children within the highest category of CHLS had an average lower BMI. Those exposed to tobacco smoke at 4 years of age (43.8%) tended to have a lower CHLS. About half of the families (48.3%) were of maternal manual laboring classes (classes IV–VI), which tended to have a lower CHLS (Table 1).

With regard to lifestyle, 45.6% reported a high adherence to the Mediterranean diet, 84.2% had more than 10 h of sleep a night, 40.9% reported less than 1 h daily of television watching, and 42.24% reported more than the median physical activity value for their respective cohort (Table 2).

Higher adherence to the Mediterranean diet and sleeping more than 10 h a day showed no significant associations with MSCA after controlling for confounders. Higher levels of physical activity were significantly and negatively associated with MSCA, which persisted after adjustment (β: −1.5, 95% CI: −2.8, −0.2). Less than 1 h a day of screen time was also significantly negatively associated with MSCA after adjustment (β: −1.8, 95% CI: −3.2, −0.5). (Table 2). The child healthy lifestyle (CHLS) score had a non-significant association with MSCA in crude and adjusted models (Table 3). Treating the CHLS score as a continuous variable, CHLS was not associated with MSCA in the adjusted model (1-point increment = −0.5, 95% CI: −1.2, 0.2).

Since major changes to the coefficients were seen after the inclusion of social class, we stratified by manual and non-manual working class. Children of the non-manual working class were reported to watch less television, do less physical activity, and were more likely to adhere to a healthy Mediterranean diet. There were no significant associations between the CHLS and MSCA observed for participants in the manual or non-manual working classes. The unexpected inverse association between healthy lifestyle factors and MSCA score was more pronounced, although non-significant, in the manual working class (Table 4).

Sensitivity analysis stratifying by cohort and sex did not show different results. Further adjustments by maternal education, maternal verbal IQ proxy, and BMI z-score at 4 years old did not significantly attenuate the estimates (data not shown). Excluding children with attention deficit/hyperactivity disorder symptoms (*n* = 87) did not change the negative association between physical activity and MSCA (data not shown). An analysis of missing data found that excluded children were more likely to be obese and exposed to tobacco smoking. In relation to parental variables, excluded children were more likely to have mothers and fathers who were of the manual working class, with no education or only primary school education (data not shown in tables). This leaves the children in this analysis who are at a healthier weight, less likely to be exposed to tobacco smoke, and in families of a higher class with more education than the population of the cohort as a whole.

## 4. Discussion

A healthier lifestyle assessed as a combined score of diet, sleep, screen viewing, and physical activity did not have any association with MSCA in a cross-sectional study of 4-year-old children. The lifestyle score was constructed to reflect an underlying construct of “healthy behaviors”, as well as to be easily comparable and reproducible in other studies. In order to do this, it was not weighted to account for the different amounts of effect that each component had on the outcome, which limits the specificity of the composite score in relation to the outcome variable.

Contrary to our expectations, more physical activity and less screen time were significantly and negatively associated with MSCA in the adjusted models. The unexpected inverse association for the composite score was stronger, although not significant in manual working class families, suggesting that this association was driven by reported behaviors in the manual working class families.

In relation to dietary factors and neuropsychological development, we found no association with the reported Mediterranean dietary score and the MSCA in this population of pre-school children. Previous research had found that, while healthy dietary patterns are better for neurocognitive development, the Mediterranean dietary score itself was only found to be beneficial in older children [17]. The Mediterranean diet has been associated with a slower cognitive decline in adults and few studies have looked at the Mediterranean diet and neuropsychological development in preschool age children [29]. Our lack of association suggests that the accumulation of positive dietary factors over a lifetime by the Mediterranean diet leads to better cognitive health and associations not seen in a cross-sectional analysis at pre-school age. With the aMED taking a positive factor approach to diet, negative factors such as “junk food” (refined and processed high calorie foodstuffs) and sugary drinks, with the exception of processed meats, were not considered. While the assumption is that children who have more points on the healthier dietary scale have less room in their daily intake of nutrients for “junk food”, this has not been measured in the study.

Sleep time was not associated with MSCA after adjustment. Most studies show that even small amounts of sleep restriction adversely affect emotional and cognitive functioning when studying sleep restriction as a singular event or in the short span of limited sleep for a day or a week. It is important to keep both the quality and quantity of sleep in mind when considering neurocognitive outcomes in children and to continue to support families to let children get enough sleep.

We found an unexpected negative association between less screen time and MSCA, more screen time was associated with a higher score on the MSCA. Data collection did not differentiate between types of screen viewing; the literature now differentiates between active and passive screen time viewing. Active screen time particularly includes exposure to computers and interactive video games, which are associated with higher scores on the MSCA [30]. The effect of passive screen time viewing, such as television viewing, on cognitive development is unclear, but purposefully well-designed interactive programming can be educational [31]. Other outcomes of increased screen time, such as increased sedentary time, can be harmful to children [3]. In our study, many children surpassed the current recommended 1 h a day limit [22]. The lack of differentiation in the data collection of types of media use could account for a lack of nuance in our analysis, especially if making a link to current and ever-changing screen use. It is important to continue the advocacy of less total screen time and keep up parental awareness of the types of programming and recommendations for preschool age children [22].

While the negative association between physical activity and cognitive ability is unexpected, this analysis can only conclude that the reported high physical activity and lower neuropsychological development are associated. Physical activity was reported outside of the pre-school environment for standardization. This measure reported good validity in other longitudinal studies of the population. Based on previous evidence, higher amounts of reported physical activity benefit neurodevelopment, which may be evident later in life, especially since complex neuropsychological functions do not fully develop until young adulthood [2,3,32]. It is possible that the time spent on physical activity takes away from time in educational activities that would increase MSCA scores, such as sedentary time spent reading or playing educational video games [30].

At four years old, children may be too young to have any neurodevelopmental changes linked to lifestyle. As they age, once unhealthy habits become ingrained, the accumulation of lifestyle choices possibly alters neuropsychological performance. The null results of lifestyle at four years do not assuage the concern that life choices may affect children later on.

As a cross-sectional analysis, we can only find associations and suggest further investigation. As with all observational studies, we cannot control for residual confounding effects apart from the measured variables. To our knowledge, no previous studies have reported physical activity and MSCA to associate as we found them to do [32,33,34,35,36]. It is possible that the measures of physical activity at such a young age are inaccurate, especially since preschool age children are assumed to be active [37]. Screen time in young children is a complex topic, which is only becoming more relevant. Currently, children are exposed to a variety of screens, like tablets and mobile phones, that were not accounted for in this study. The unexpected positive association with television and computer use and cognitive ability suggests that nuance in the measurement of and recommendations for screen time is necessary. Stratifications by social class, sex, and cohort also found similar results, which leads us to suspect an association not previously reported with physical activity and screen time that relates to neuropsychological development. With regard to the outcome, neuropsychological data were assessed following standard tools and methods and showed good psychometric validity [25].

## 5. Conclusions

No association between healthy lifestyle habits and neuropsychological development, as measured by the McCarthy Scales of Children’s Abilities, was found in 4-year-old children, quantified by an overall index score. Further analysis into the role of physical activity and screen time with neuropsychological development must be done to verify the unexpected results, and to consider causation. It is possible that the association between the exposure measures of lifestyle and the outcome measures of neuropsychological development is not yet applicable in children at 4 years of age since they have had exposure to a certain lifestyle. Considering this is a cross-sectional look at these behaviors and outcomes, a longitudinal measure of healthy lifestyle, particularly physical activity and screen time and the association with neuropsychological outcomes, is warranted.

## Figures and Tables

**Table 1 ijerph-17-05668-t001:** Child and parental characteristics according to the child healthy lifestyle score (CHLS) at pre-school age.

Family Characteristics		CHLS Categories				
		0 and 1	2	3	4	*p*-Value
	*n*	(*n* = 430)	(*n* = 627)	(*n* = 467)	(*n* = 126)	
Child						
Sex ^+^, %						
Female	807	44.7	48.3	52	54.8	0.080
Male	843	55.3	51.7	48	45.2	
BMI categories, %						
Normal weight	1124	66.3	69.5	67.5	69.8	0.042
Overweight	364	20.5	21.1	23.8	26.2	
Obese	162	13.3	9.4	8.8	4	
BMI in kg/m^2^, mean (SD)	1650	16.4 (1.9)	16.2 (1.6)	16.3 (1.6)	16 (1.4)	0.480
MCSA general cognitive score, mean (SD)	1650	100.5 (14.6)	100.1 (15)	101 (14.3)	101 (16.3)	0.793
Cohort ^+^, %						
Valencia	379	18.8	20.7	27.2	32.5	<0.001
Sabadell	375	15.6	23.3	26.3	31	
Asturias	402	23.3	25.2	25.1	21.4	
Gipuzkoa	494	42.3	30.8	21.4	15.1	
Exposed to tobacco smoke ^+^, %						
No	927	49.5	55.4	63.1	58.4	0.001
Yes	723	50.5	44.6	36.9	41.6	
Mother						
Social class ^+^, %						
SC I + II	398	17.7	24.1	27.8	32.5	<0.001
SC III	455	27.0	26.6	28.7	30.2	
SC IV–VI	797	55.3	49.3	43.5	37.3	
Educational level, %						
Primary or less	338	23.5	22.3	18.4	8.7	<0.001
Secondary	685	46.7	39.9	40.3	36.5	
University	627	29.8	37.8	41.3	54.8	
Maternal IQ, mean (SD)	1561	9.6 (3)	10 (2.9)	10.1 (2.8)	10.5 (3.3)	0.041
Father						
Social class ^+^, %						
SC I + II	355	16.5	20.4	24.4	33.3	0.001
SC III	299	17.2	18	19.7	15.9	
SC IV–VI	996	66.3	61.6	55.9	50.8	

BMI: Body Mass Index; SD: Standard Deviation; MSCA: McCarthy Scales of Children’s Abilities; IQ: Intelligence Quotient. SC: Social class, ^+^ Considered a confounder in the model because of statistical significance or a priori. Overweight is defined as a BMI at or above the 85th percentile and below the 95th percentile for children of the same age and sex; obesity is defined as a BMI at or above the 95th percentile for children of the same age and sex.

**Table 2 ijerph-17-05668-t002:** Association between the individual lifestyle factors and the general cognitive McCarthy score as a continuous variable, at 4 years old (*n* = 1650).

Lifestyle Factors and MSCA		Model 1	Model 2	Model 3
Lifestyle Factors	*n* (%)	β (95% CI)	β (95% CI)	β (95% CI)
High-quality diet (aMED)	752 (45.58)	1.2 (−0.2, 2.6)	0.9 (−0.5, 2.2)	0.9 (−0.5, 2.3)
Sleeping > 10 h a day	1390 (84.24)	**2.2 (0.3, 4.1)**	1.3 (−0.5, 3.1)	1.6 (−0.2, 3.4)
TV/screen watching < 1 h a day	674 (40.85)	−0.4 (−1.8, 1.0)	**−1.5 (−2.8, −0.1)**	**−1.8 (−3.2, −0.5)**
Physical Activity > cohort-specific median METs	697 (42.24)	**−1.8 (−3.2, −0.4)**	**−1.5 (−2.8, −0.2)**	**−1.5 (−2.8, −0.2)**

aMED: Alternative Mediterranean Diet Score; METS: Metabolic Equivalents. Model 1: Adjusted for test quality, age at McCarthy test, cohort, and sex. Model 2: Model 1 + exposure to cigarette smoke, maternal and paternal social class (and calories in aMED). Model 3: Model 2 and mutually adjusted for other lifestyle factors. Bold indicates statistical significance.

**Table 3 ijerph-17-05668-t003:** Association between the child healthy lifestyle score (CHLS) and McCarthy general cognitive score at 4 years old (*n* = 1650).

Lifestyle Score and MSCA	Model 1 (Minimally Adjusted)	Model 2 (Fully Adjusted)
	β (95% CI)	β (95% CI)
CHLS categories		
0 and 1	Ref.	Ref.
2	−0.5 (−2.2, 1.3)	−1.0 (−2.7, 0.6)
3	0.2 (−1.7, 2.1)	−1.0 (−2.8, 0.8)
4	−0.1 (−2.9, 2.7)	−2.0 (−4.8, 0.8)
*p* for trend	0.845	0.152

Model 1: Adjusted for test quality, age at McCarthy test, cohort, and sex. Model 2: Model 1 + exposure to cigarette smoke, maternal and paternal social economic class.

**Table 4 ijerph-17-05668-t004:** Association between child healthy lifestyle score (CHLS) and McCarthy general cognitive score at 4 years old, stratified by maternal manual and non-manual working class (*n* = 1650).

Lifestyle Score and MCSA	Non-Manual Working Class (I–III) (*n* = 853)	Manual Working Class (IV–VI) (*n* = 797)	*p* for Interaction
	β (95% CI)	β (95% CI)	
CHLS categories			
0 and 1	Ref.	Ref.	
2	−0.5 (−2.9, 1.9)	−1.4 (−3.8, 1.0)	0.726
3	−0.8 (−3.3, 1.7)	−1.0 (−3.7, 1.7)	
4	−0.8 (−4.3, 2.7)	−4.0 (−8.5, 0.5)	
*p* for trend	0.548	0.151	

Model adjusted for test quality, age at McCarthy test, cohort, sex, exposure to cigarette smoke, paternal social economic class.

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
