# Peer review of "Association of Lifestyle Factors and Neuropsychological Development of 4-Year-Old Children"

_ijerph, 2020, doi:10.3390/ijerph17165668_

Round 1

Reviewer 1 Report

I appreciate the authors efforts to provide additional content.  However, these additions do not address my initial concern that the authors present data from one time point. Doing so very much constrains what the authors can say concerning how children's early exposure to specific factors/conditions impacts  their cognitive development at age 4.

Author Response

I appreciate the authors efforts to provide additional content.  However,
these additions do not address my initial concern that the authors present
data from one time point. Doing so very much constrains what the authors can
say concerning how children's early exposure to specific factors/conditions
impacts  their cognitive development at age 4.

We would like to thank the reviewer for appreciating our effort. We agree that cross-sectional studies are limited, but nevertheless they are still used and frequently published, for hypothesis generation. It is possible that the study design limited the interpretation of our findings, but these findings deserve being published in order to encourage further research in this area, and also to avoid publication bias of null findings. We hope future research can be performed using longitudinal data in order to confirm or refuse our findings. 

Reviewer 2 Report

Authors made great efforts in improving their original manuscript.

I think this version could be acceptable for publication.

Author Response

Authors made great efforts in improving their original manuscript. I think
this version could be acceptable for publication.

We would like to thank the reviewer for appreciating our effort.

Reviewer 3 Report

The authors have addressed most of the previous concerns.  My remaining concern relates to the correlations between the variable included in the index.  Upon inspection of the correlation matrix, TV viewing is highly correlated (.77) with aMed diet and moderately correlated with sleep (as expected).  Also, physical activity is moderately correlated with aMed diet.  The authors can leave the index as is, but really need to stress these in the results and discussion, as these behavioral variables together may reflect an underlying construct of 'healthy behaviors'.

Author Response

The authors have addressed most of the previous concerns.  My remaining
concern relates to the correlations between the variable included in the
index.  Upon inspection of the correlation matrix, TV viewing is highly
correlated (.77) with aMed diet and moderately correlated with sleep (as
expected).  Also, physical activity is moderately correlated with aMed diet.
The authors can leave the index as is, but really need to stress these in
the results and discussion, as these behavioral variables together may
reflect an underlying construct of 'healthy behaviors'.

We would like to thank the reviewer for appreciating our effort in addressing your concerns. There was a typing error in the previous version, and the correlation coefficient between TV viewing and aMed diet is 0.1799, non-statistically significant (see table below). Low-to-moderate correlation between components is observed, and this is expected. Indeed, the aim of creating and index-score of healthy lifestyle factors, is to account in part for the correlation between its components and reflect an underlying construct of “healthy behaviours”, as pointed out by the reviewer. We have highlighted this in the discussion, as suggested by the reviewer.

Supplementary table. Estimated correlation between lifestyle factors

Lifestyle factors

 1. TV (< 7 hr/week)

2. Sleep (>70 hrs per week)

3. Physical Activity (> median METs of cohort)

4. aMEDKID (> 3 points)

1. TV

2 Sleep

0.1599*

3 PA

-0.0595

0.0104

4 aMed diet

01799

0.1386*

0.0305

VIF (collinearity)

1.02

1.02

1.0

1.02

*Statistically significant (p =< 0.001)

 Tetrachoric correlation,

 Binary variables made by recommended value or median specific (PA)

Reviewer 4 Report

The revised manuscript is improved and all comments have been satisfactorily addressed.

Author Response

The revised manuscript is improved and all comments have been satisfactorily
addressed.

We would like to thank the reviewer for appreciating our effort.

This manuscript is a resubmission of an earlier submission. The following is a list of the peer review reports and author responses from that submission.

Round 1

Reviewer 1 Report

The authors of the article entitled “Association of Lifestyle Factors and Neuropsychological Development of 4-Year-Old Children” present a well-structured, quality work with an appropriate sample and analysis of pertinent data.
From my point of view, I think the only aspect that needs to be improved is the introduction, since it is somewhat reduced (only two paragraphs). I encourage the authors to develop the theoretical framework in greater depth, providing concrete data from previous studies, and the main theoretical foundations and hypotheses that support their work.

Reviewer 2 Report

The authors examine the relationships between four primary lifestyle factors identified as Mediterranean diet (which needs to be clear for all readers), television screen time (which may be inappropriate given other screens to which children have access), sleep time, and physical activity among 4 year olds and their cognitive development on the MSCA (for which we do not learn the internal consistencies for scales among the present sample).  Why these factors were selected is not clear as one learns little about what exactly is being measured in the larger INMA study. Given that only one age group was examined, this study cannot be referred to as a cross-sectional in nature.

Regardless, I do have questions as to why the authors included only one time point in which to examine the extent to which the composite of their lifestyle factors was linked to cognitive development on only one measure of that development.  Given that ages 3-5 is noted as a time for the development of executive functioning (EF), one might have expected a measure of EF. Presumably such a measure was not included in the larger study from which data for this manuscript was drawn. Regardless, data gathering at more regular time points throughout the early childhood period would have been more informative as to what early exposure factors, as selected by the investigators, were linked to cognitive development during early childhood.  Ideally, the researchers would have conducted a cross-lagged panel analysis.

Independent of my concerns, the authors failed to find significant results linking their composite lifestyle score to cognitive performance. The negative correlation between physical activity and cognitive performance is potentially interesting but not sufficiently so to override other major flaws with the study.

I do see the authors' work as an interesting start to a larger longitudinal investigation of differential contributors to children's cognitive development.  I do not see the work as presented as publishable in its own right.

Reviewer 3 Report

O´ Connor et al., performed a cross-sectional study to analyze how (individual or combined) lifestyle factors such as diet, sleep, screen viewing and physical activity can affect the neuropsychological development of 4 years old children.

The topic of research is interesting and the sample size included in the study is considerable. However, there are some aspects in methods that need to be better explained and some incongruences when comparing results and table 1 that the authors must revise and correct. These are addressed below, section by section:

UNDER METHODS:

  • The authors state: “Children were assessed through a series of tests” (line 88). Please, detail which kind of tests were used.
  • Lines 89-91: the authors should also better explain which exclusion criteria they used.

UNDER RESULTS:

Under abstract (line 33) and methods (line 91) paragraphs the authors state that they performed the study on a population including 1650 children but Table 1 shows some incongruences:

  • In “CHLS categories” row, under “All” column the number of subjects is 1434
  • In “Sex” row, number of female + male subjects sums 1555
  • Under “weight categories” row, number of normal weight + overweight + obese subjects sums 1426
  • Under “exposed to tobacco categories” row, number of Yes + No subjects sums 1431

More under results:

  • The authors state (line 173) that 44,4% of children exposed to tobacco tended to have lower CHLS, but the percentage reflected in table 1 is 43,3.

Please revise and correct all these incongruences.

MINOR CONCERNS:

  • Some typographical errors are present in the text. Please carefully revise the whole manuscript. For example: (Line 174) please change “(classes IV-V)” with (classes IV-VI).

Reviewer 4 Report

I have a concern if this manuscript is appropritate for IJERPH. 

Maybe Behav. Sci. could better fit.

However, the lack of association the authors found could be related to the very young age of the enrolled children. Any possibility to perform the same experimental design on other children of different age?

Reviewer 5 Report

This paper takes advantage of the well designed INMA cohort study in Spain, and uses 4 of the component cohorts to examine associations between 4 very important life style variables measured at age 4 years and the cross sectional associations with cognition measured by the McCarthy Scales.  The results are somewhat surprising with the healthy lifestyle variables either having no association with total MSCA scores or inverse associations, particularly among the manual classes.  

  1. How did participants in the final sample for this analysis differ from the cohort of 2644?
  2.  (Major) The healthy lifestyle index really weighs each component equally, when they may not make equal contributions to the outcome variables.  Please comment on this potential limitation. 
  3.  Not surprisingly, it seems like the four variables which contribute to the CHLS are correlated.  Please provide these correlations as well as the correlations.
  4.  If indeed the four components are highly correlated, it may make sense to construct a 'mixtures' analysis, using a method such as weighted quantile scores, or Bayesian Kernel  Machine Regression, which also accounts for interactions between the components.
  5. The investigators may also wish to present in the supplement a DAG which would inform how they selected the control variables.
  6.  As the authors indicate in the discussion, the meaning of a  Mediterranean diet in children is not clear.  Perhaps rethinking the diet variable to incorporate fast food or 'junk' food consumption would be a good idea as well. 

In summary, this paper needs a bit of analytical work to sort out the surprising results. 

  1.